# Feasibility and ethics of using data from the Scottish newborn blood spot archive for research

Sarah Cunningham-Burley[1,14✉], Daniel L. McCartney [2,14], Archie Campbell [2,14], Robin Flaig [3,14], Clare E. L. Orange[4,14], Carol Porteous[1,12], Mhairi Aitken[1,13], Ciaran Mulholland[5], Sara Davidson[5], Selena M. McCafferty[4], Lee Murphy [6], Nicola Wrobel[6], Sarah McCafferty[6], Karen Wallace[7], David StClair[8], Shona Kerr [9], Caroline Hayward [9], Andrew M. McIntosh [10], Cathie Sudlow[3], Riccardo E. Marioni [2], Jill Pell[11], Zosia Miedzybrodzka[7] & David J. Porteous [2✉]

**Abstract**

**Background** Newborn heel prick blood spots are routinely used to screen for inborn errors of metabolism and life-limiting inherited disorders. The potential value of secondary data from newborn blood spot archives merits ethical consideration and assessment of feasibility for public benefit. Early life exposures and behaviours set health trajectories in childhood and later life. The newborn blood spot is potentially well placed to create an unbiased and cost-effective population-level retrospective birth cohort study. Scotland has retained newborn blood spots for all children born since 1965, around 3 million in total. However, a moratorium on research access is currently in place, pending public consultation.

**Methods** We conducted a Citizens' Jury as a first step to explore whether research use of newborn blood spots was in the public interest. We also assessed the feasibility and value of extracting research data from dried blood spots for predictive medicine.

**Results** Jurors delivered an agreed verdict that conditional research access to the newborn blood spots was in the public interest. The Chief Medical Officer for Scotland authorised restricted lifting of the current research moratorium to allow a feasibility study. Newborn blood spots from consented Generation Scotland volunteers were retrieved and their potential for both epidemiological and biological research demonstrated.

**Conclusions** Through the Citizens' Jury, we have begun to identify under what conditions, if any, should researchers in Scotland be granted access to the archive. Through the feasibility study, we have demonstrated the potential value of research access for health data science and predictive medicine.

**Plain language summary**

In Scotland, babies are checked for conditions treatable early on that might otherwise be serious or fatal. These checks are done on dried heel prick blood spots that are then stored. We investigated if they might help research into medical conditions that develop later in life. To answer this, we set up a Citizens Jury, a small group of people representative of the area, which met over a few days to consider the issue and decide on it. They agreed it could be valuable but wanted reassurance about what research would be conducted and how it would be monitored and controlled. Using samples from research volunteers, we have extracted medically relevant information from samples that had been stored for over 30 years. This suggests a wider public consultation should be carried out to agree on the conditions of future research.

---

[1] Centre for Biomedicine, Self and Society, Usher Institute, University of Edinburgh, 23 Buccleuch Place, Edinburgh EH8 9LN, UK. [2] Centre for Genomic and Experimental Medicine, Institute of Genetics and Cancer, University of Edinburgh, Western General Hospital, Edinburgh EH4 2XU, UK. [3] Centre for Medical Informatics, Usher Institute, University of Edinburgh, Nine, Edinburgh Bioquarter, 9 Little France Road, Edinburgh EH16 4UX, UK. [4] NHS GGC Biorepository, Level 3, Laboratory Medicine Building, Queen Elizabeth University Hospital, 1345 Govan Road, Glasgow G51 4TY, UK. [5] Edinburgh Clinical Research Facility, University of Edinburgh, Western General Hospital, Edinburgh EH4 2XU, UK. [6] Ipsos MORI Scotland, Links House, 15 Links Pl, Edinburgh EH6 7EZ, UK. [7] Medical Genetics, Room 2:041, School of Medicine, Medical Sciences and Nutrition, Polwarth Building, Foresterhill, Aberdeen AB25 2ZD, UK. [8] School of Medicine, Medical Sciences and Nutrition, Foresterhill Health Campus, Foresterhill Rd, Aberdeen AB25 2ZN, UK. [9] MRC Human Genetics Unit, Institute of Genetics and Cancer, University of Edinburgh, Western General Hospital, Edinburgh EH4 2XU, UK. [10] Centre for Clinical Brain Sciences, Division of Psychiatry, University of Edinburgh, Royal Edinburgh Hospital, Edinburgh EH10 5HF, UK. [11] Institute of Health and Wellbeing, University of Glasgow, Glasgow G12 8RZ, UK. [12]Present address: Edinburgh Clinical Research Facility, University of Edinburgh, Western General Hospital, Edinburgh EH4 2XU, UK. [13]Present address: The Alan Turing Institute, British Library, 96 Euston Road, London NW1 2DB, UK. [14]These authors contributed equally: Sarah Cunningham-Burley, Daniel L. McCartney, Archie Campbell, Robin Flaig, Clare E. L. Orange. ✉email: Sarah.C.Burley@ed.ac.uk; david.porteous@ed.ac.uk

Longitudinal population health studies with extensive record linkage and deep genomic annotation have the potential to realise the objectives of preventative and predictive medicine. Ideally, these studies should cover the full life course. Epidemiological and biological analyses of newborn blood spots have the potential to anchor longitudinal studies at birth. The Danish Newborn Screening Biobank (DNSB) has shown that nucleic acid, metabolic and protein assays are feasible on archived newborn dried blood spots[1]. Denmark is currently the only country in the world where nation-level linkage of such secondary data to health records is approved[1]. However, issues of consent and sensitivity over genetic studies first need careful consideration. A Scottish Government commissioned review had previously considered the ethical, legal and social implications of research access to the Guthrie Card archive[2]. In Scotland, the Guthrie card archive (henceforth referred to as the Scottish newborn blood spot archive) is considered part of the medical record and subject to the provisions of the Data Protection Act 1998. The report recommended that the archive should be treated as both tissue and data in legal governance. The report also recommended wider public engagement on the issues. There are many ongoing research studies in Scotland that have obtained consent from study participants for NHS health record linkage. Others have explicit consent to use participants' newborn blood spots. Nevertheless, a research study moratorium remains in place. The purpose of this study was to provide supporting evidence for a planned Scottish Government commissioned public consultation.

Here we report the verdict of a Citizens Jury that conditional research access to the newborn blood spots was in the public interest. We show that newborn blood spots from consented Generation Scotland volunteers that have been stored for over 30 years can be retrieved from the archive. We demonstrate that DNA can be reliably extracted from these blood spots for genetic and epigenetic analysis.

## Methods

**Citizens' Jury**. In June 2017, we brought together a small, diverse group of citizens to address the question: "Would research access to the Guthrie Card heel prick blood tests be in the public interest, and, if so, under what conditions?" A Citizens' Jury is a well-established method to enable public participation in policy making, allowing informed deliberation on an issue and the provision of recommendations. Our Citizens' Jury followed best practice for such deliberative public engagement[3]. First, we convened a steering group to provide oversight of the materials prepared for the Jurors and to identify a range of expert witnesses to give evidence. Next, we consulted a Patient Participatory Involvement and Engagement (PPIE) panel to review these materials. The academic researchers did not involve themselves directly in the Jury process, other than to provide evidence or observe. The recruitment, facilitation and analysis were conducted by Ipsos MORI, Scotland, to preserve neutrality. Jurors met together for two day-long sittings to hear evidence (neutral, for and against), deliberate and reach conclusions.

Using a quota sample, a representative pool of the adult public was recruited in terms of sex, age, working status and social grade. Additional quotas were set to ensure sufficient representation of people with children under the age of five, and people with a family history of a medical condition. An attitudinal quota was set to ensure inclusion of people with varied levels of trust in public, private and third sector organisations, as previous research has found this to be a significant factor underpinning views of data sharing and use[4]. A total of 20 were recruited of whom 19 participated on day one, and 18 returned for the second

day a week later. Jurors were given monetary recompense for taking part in each sitting.

Day One started with a warmup session sharing their views on health research and health-relevant information, followed by evidence from various experts to stimulate discussion of issues around research use of the newborn blood spots. Day Two included further expert witnesses but more time for Jurors own deliberations and to arrive at a conclusion on the key question. Facilitation tools, such as speed dating techniques, meant that all Jurors could express and reflect on their own views as well as those of others, building up to group and plenary deliberations. The first day focused on more general discussion and information sharing; the second day involved detailed deliberation of the key question and delivering of the verdict. Both days were audio recorded and transcribed for subsequent analysis. A short questionnaire was administered at the end of each day to gauge individual level views and thinking.

Background information was provided by authors SCB and DJP during the morning of Day One. Over the 2 days, six witnesses were called to give evidence and answer Jurors' questions. These comprised health care professionals and scientists, a Caldicott Guardian (a person responsible for protecting the confidentiality of people's health and care information and making sure it is used properly) and a Genewatch spokesperson (representing not-for-profit groups that monitor developments in DNA technologies). Jurors also had access to a short video and other written information about comparative deliberations and policy in California and Denmark. Having heard on Day One some of the health research opportunities uniquely possible if access were granted, an introduction to some of the social, ethical and legal issues, and how the NHS protects privacy, Jurors heard more opposing and critical views on Day Two, with one witness cautioning against allowing research access and another the importance of not compromising the newborn screening programme itself.

### Newborn blood spot documentation and sampling

*Epidemiological feasibility.* The Scottish newborn blood spots archive is stored in around 900 boxes each containing circa 3000 cards in a single secure location under the authority of the Director of the Scottish Newborn Screening laboratory and custodianship of the NHS Research Scotland Greater Glasgow and Clyde Biorepository. A unified digital record is in place from 2000 onwards, but the content and consistency of information available from older cards was not known at the outset. The design and information content of the newborn blood spots was known to have varied over time, but not documented. We were aware that some cards had suffered water damage prior to assembling the nation-wide archive. We also know that for a period of time some cards had been autoclaved before storage. For the vast majority, the newborn blood spots have simply been stored at room temperature. The unified newborn blood spot archive management database has limited, high-level summary information on the contents of each box.

We obtained permission from the Caldicott Guardians of NHS Research Scotland Greater Glasgow and Clyde and of Tayside to retrieve representative boxes from the NHS Research Scotland secure archive. Permission was given for (a) examination and documentation of a sub-sample from each box to provide a snapshot of the information attached that might be required from the purposes of linkage to other routine health records and (b) sampling of cards corresponding to consented members of Generation Scotland.

Thirty boxes representing each decade from 1965 to 1999 were retrieved and examined by NHS Scotland staff at the NHS

**Table 1 Newborn blood spots blood spot DNA extraction and Sanger sequencing.**

| Time period | Newborn blood spots sampled | Successful[a] DNA extractions | Extractions that failed DNA sequencing | Mean yield of DNA extracted (μg) | Net yield for analysis |
|---|---|---|---|---|---|
| 1965–1974 | 17 | 15 | 0 | 16.7 | 88% |
| 1975–1984 | 32 | 31 | 1 | 19.5 | 94% |
| 1985–1994 | 17 | 16 | 1 | 17.3 | 88% |
| 1995–2004 | 18 | 17 | 0 | 23.1 | 94% |
| 2005–2019 | 16 | 16 | 1 | 22.9 | 94% |
| 2010–2013 | 36 | 36 | 0 | 19.7 | 100% |
| Sum | 136 | 131 | 3 | 19.9 | 94% |

[a]In 5/136 (3.8%) the extraction process failed to recover measurable quantities of DNA using the methods described.

Research Scotland Greater Glasgow and Clyde Biorepository. Only summary information was shared with the rest of the study team.

The organisation and information content by box varied over time (Supplementary Fig. 1). In some boxes there were unlabelled bundles of cards (Supplementary Fig. 1D), but most had 4 labelled foolscap sub-boxes (Supplementary Fig. 1B) with multiple, date-labelled bundles of cards (Supplementary Fig. 1C).

We drew up a list of potential information that the newborn blood spots might carry from which to judge the feasibility of conducting epidemiological studies by linkage to NHS Scotland routine medical record and potentially additional consented data from research subjects.

For each box, we collected the following information: Box ID; Area Health board; Hospital; Type of card; Date of test; Child forename; Child surname; presence or absence of Community Health Index identifier. No personal information was recorded. Each box took 2 members of staff working in tandem ~2 h to document.

Next, circa 1 in 100 cards from each box were examined in detail and the presence or absence of the following features documented: Child DOB; Additional comments on card; Number of blood spots; Size of spots; Mother's CHI; Mother's forename, surname and birth name; Mother's date of birth; Address; Postcode; Whether the cards had been autoclaved prior to archiving; Any other comments.

Cards from 1965 had very little information on them and in many cases did not even record the sex of the baby. Information content increased progressively over time. By the 1990s, the sex of the child and home address were generally recorded.

**Retrieval of cards from consented volunteers**. Over 24,000 Generation Scotland (GS) volunteers were recruited as adults between 2006 and 2011[5]. All were born before 1993, before the digital recording of newborn blood spots began. Consent for linkage to medical records was optional but was given by 98% of volunteers. They were asked to give information about their place of birth (country and council area). A total of 8703 volunteers with linkage consent were born in Glasgow or Tayside area health boards between 1965 and 1992. The set of 30 boxes retrieved and documented for epidemiological purposes were selected on the basis that (a) Greater Glasgow and Tayside were the regions for which we had Caldicott Guardian approval and (b) they were expected to include bundles from Generation Scotland participants as the majority came for these regions. A list of names, birthplace and date of birth was extracted from the GS database and sent to the NHS Greater Glasgow and Clyde Biorepository to look for matching cards. Pseudonymous ids were added to the list so that any samples from matching cards could be labelled and linked back to the GS database after genotyping. Ninety-two matching cards were identified amongst newborn blood spots

from Tayside. Of these, 58 were usable for punching having fulfilled the prerequisite of leaving one spot intact (Supplementary Fig. 2). Six to ten punches were taken from each card, placed in vials labelled with a pseudonymous ID for matching to the samples donated at baseline by each Generation Scotland volunteer, and couriered to the Edinburgh Clinical Research Laboratory Genetics Laboratory for DNA analysis.

**Biological feasibility**

*Genetic analysis of de-identified cards*. Unlike the Danish Newborn Screening Biobank (DNSB), the Scottish newborn blood spot archive comprises a variety of paper types and storage conditions, particularly for older cards. To establish the effect this might have on the recovery of analysable DNA, a pilot study in 2012–2014 was undertaken on a de-identified set of 136 newborn blood spots dated from 1965 to 2012 (Table 1). The study was mandated by the Scottish Chief Scientist Office, following a favourable opinion from the Scottish Legal Office and North of Scotland Research Ethics Committee. De-identified cards were provided by the Scottish National Dried blood spot collection, Biochemical Genetics Laboratory, Duncan Guthrie Institute, Greater Glasgow Health Authority, Yorkhill, Glasgow. DNA was extracted from 3 mm punches using the Sigma ENA kit. Yields varied from sample to sample, but there was no significant effect of date of birth and sufficient material was obtained for Sanger DNA or exome sequencing in 94% of samples (Table 1). Exome sequencing used the Ion AmpliSeq exome kit run on IonTorrent Proton sequencer. Data analysis and variant calling used the IonReporter IonExpress variant caller, 42–45 million mapped reads, 94.2–94.7% on target, mean depth 122–130 reads per sample. Thirty one of 32 runs met standard QC criteria for variant analysis.

*Sampling of cards*. Of the 92 newborn blood spots matched to GS volunteers, 58 (63%) had sufficient dried blood spot material remaining to take 3 mm diameter punch samples, while leaving at least one spot intact. These 58 were from Generation Scotland research volunteers born between 1983 and 1989 (i.e. 32–38 years between collection and profiling). DNA was extracted from between 6 to 8 blood spot punches using the QIAamp DNA Investigator Kit (Qiagen; cat. no. 56504), following the manufacturer's instructions. The concentration of the DNA samples was measured using a Qubit 2.0 fluorometer and the Qubit dsDNA HS assay (Thermo Fisher; cat. no. Q32854). Total yield isolated was between 196 and 1177 ng of DNA. Up to 500 ng DNA (range 160–500 ng) underwent bisulfite conversion (Zymo EZ-96). DNA methylation was profiled using the Infinium HumanMethylationEPIC v1.0 BeadChip (Illumina Inc.; cat. no. WG-317-1001), according to the manufacturer's protocol (in batches of 8 samples, 56 assayed of the 58 samples processed).

Arrays were scanned on an iScan and analysed using GenomeStudio v2011.1.

*DNA methylation analysis and statistical methods*
DNA methylation quality control: DNAm profiles were obtained from the 56 individuals using the Illumina MethylationEPIC beadchip, measuring ~850,000 CpGs across the genome. Quality control measures were performed, removing probes with high detection *p*-values (>0.05) in >5% of samples ($N = 52,375$), or a beadcount <3 in more than 5% of samples ($N = 5038$). Three samples were removed for having >5% of sites with a detection *p*-value >0.05. In addition to these standard quality control measures, additional checks were performed to ensure newborn blood spots and baseline samples matched with regard to predicted sex and genotype (Supplementary Information Methods, Supplementary Figs. 3 and 4 and Supplementary Table 1). Quality control and analysis code have been deposited in a public repository[6]. To access Generation Scotland data, including the data derived in the feasibility study described here, please go to www.ed.ac.uk/generation-scotland/for-researchers/access.

Sample checks: Confirmatory analyses were performed using newborn blood spots DNA methylation data to ensure predicted sex (using X-chromosome data) and genotype (using "rs" control probes on the EPIC array) were consistent with peripheral blood-based genotyping and DNA methylation data on samples collected at baseline recruitment (2006–2011) (Supplementary Table 1). Detailed information on sample checks is presented in Supplementary Information Methods and Supplementary Figs. 3 and 4.

Smoking: An individual's smoking status can be reliably predicted using composite DNA methylation-derived smoking scores, and effects have also been observed in the offspring of mothers who smoked during pregnancy[7]. Moreover, information from a single probe in the aryl-hydrocarbon receptor repressor gene (*AHRR*; cg05575921) can serve as a robust marker of smoking, with lower DNA methylation levels associating with current smoker status. Maternal smoking status at the time of sample collection was derived from smoking status at GS baseline, and the "years stopped" variable for former smokers, where both mother and baby are in GS. DNA methylation-based estimates of smoking status were obtained, using previously validated methods[8]. A composite score for smoking status (EpiSmokEr) was obtained using Guthrie sample DNAm data and, along with cg05575921 DNA methylation levels, was plotted against maternal smoking status at the time of sampling (Fig. 1). Consistent with previous literature, a higher overall value was observed for the EpiSmokEr score in the offspring of current smokers whereas a lower overall value was observed for the offspring of never smokers, supporting an association at the population level (Fig. 1a; ever smoker $\beta = 0.78$; sex-adjusted linear regression $P = 0.026$). DNA methylation levels at cg05575921 were also consistent with the literature, with lower overall levels in the offspring of current smokers relative to never smokers (Fig. 1b; ever smoker $\beta = -0.72$; sex-adjusted linear regression $P = 0.05$).

**Research ethics**. The original Sanger DNA and exome sequencing study was mandated by the Scottish Chief Scientist Office, following a favourable opinion from the Scottish Legal Office and the North of Scotland Research Ethics Committee, REC ref. 11/ns.0014. A letter approving the inspection and documentation of newborn blood spots and selective sampling of GS cards for methylation analysis was provided by the Chief Medical Officer for Scotland on 4 September, 2019. The Caldicott Guardians of

NHS Greater Glasgow and Clyde and NHS Tayside granted approval on 30 January 2020 and 3 March 2020, respectively. Volunteers for Generation Scotland gave informed consent at the time of recruitment for biological studies, including genetic studies, on their biological samples and for linkage to medical records. A substantial amendment to the Research Tissue Bank approval for Generation Scotland to cover the feasibility study was submitted to the East of Scotland Research Ethics Committee and approved on 13 March 2020.

The Citizens Jury followed INVOLVE guidelines and was conducted by the polling organisation, Ipsos MORI, on behalf of the University of Edinburgh research team. This work was carried out in accordance with the requirements of the international quality standard for Market Research, ISO 20252:2012, and with the Ipsos MORI Terms and Conditions which can be found at http://www.ipsos-mori.com/terms. Ipsos MORI conducted their own internal ethical review through their ethical review team. The Ipsos MORI Project Director (CM) was then responsible for ensuring that the research materials (recruitment screener, participant information sheets, discussion guides) were clear and met the ethics principles on informed consent, right to refuse, principles of anonymity and confidentiality. No sensitive information was collected. Materials were saved in a secure folder with access restricted to the Ipsos MORI team (CM, SD). After completion, all personal information (participant names, contacts details, recordings and transcripts) were securely destroyed using Ipsos MORI digital shredding software.

**Reporting summary**. Further information on research design is available in the Nature Research Reporting Summary linked to this article.

## Results
**Citizens' Jury**. In June 2017, we conducted a 2-day Citizens' Jury in partnership with the polling organisation, IPSOS MORI following INVOLVE guidelines[3]. The question posed was "Would research access to the Guthrie Card heel prick blood tests be in the public interest?" The responses were systemically analysed to identify substantive themes and key messages ('Methods'; Table 2; Supplementary Table 2; Full report described in Supplementary Note 1). Overall, Jurors were very positive about the importance of health research and a willingness to allow their medical records to be used for such purposes. However, reservations were expressed about possible involvement of commercial actors and also about data security. After hearing from expert witnesses, the Jurors were overwhelmingly positive in terms of expressing conditional support for the use of newborn blood spots in research. Jurors felt that there was a 'clear public interest' case. This case rested on the potential advancements in the identification and/or prevention of disease, development of new treatments and ultimately, improved population health. For some Jurors, it seemed inherently wrong and wasteful that these blood spots could not to be used for research. Nonetheless, questions and concerns were raised around three main themes: data protection and security; control and oversight; and the possibility of overstretching the allowed uses. Although Jurors were reassured by how NHS data were protected, they were acutely aware that mistakes do happen and that any leak or misuse could be serious. Some concern was expressed about whether an individual would wish to know they had been identified as at risk of a health-related condition and the obligations of researchers in this regard. A small number of Jurors expressed concern about the risk of the use of the blood spots by, for example, insurance companies, and the need to restrict use to bona fide health research. A small number of Jurors expressed concern that there would be

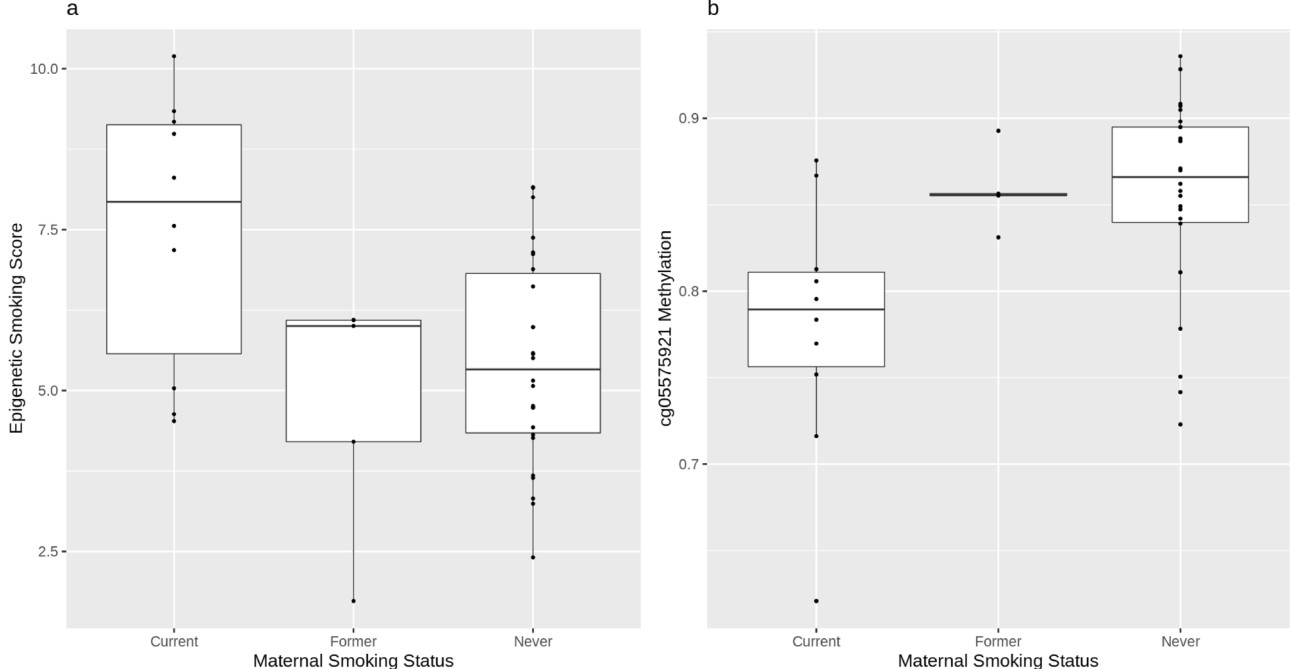

**Fig. 1 EpiSmokEr score and *AHRR* DNAm plotted against maternal smoking status.** Methylation-derived smoking scores from newborn blood spot DNA (y-axis) plotted against maternal smoking status (current, former, never) at time of birth ($N_{CurrentSmokers} = 10$; $N_{FormerSmokers} = 5$; $N_{NeverSmokers} = 26$). Results are shown for the EpiSmokEr score, a composite measure comprised of multiple CpG sites (**a**), and DNA methylation levels at a single CpG (cg05575921) in the *AHRR* gene (**b**). Upper and lower hinges correspond to the upper and lower quartiles, respectively. Whiskers extend to data points as far as 1.5 times the interquartile range. Outlying data points are defined as those beyond the whiskers. Thick horizontal lines represent the median.

---

**Table 2 Citizens Jury conditions and recommendations regarding research access.**

Condition 1: The ultimate purpose of the research should be to advance medical research in ways that could potentially benefit public health.
Condition 2: Any organisation requesting access to the data should provide a clear rationale for their research, which should be approved by an ethics committee. Access should be granted solely for the use outlined in their research request.
Condition 3: There should be appropriate regulation and oversight of the process by an independent body—that includes an ethics committee—with the authority to issue sanctions for misuse.
Condition 4: Consent should be based on an opt-out system, with the option to opt-out within one year of the child being born, and again when the child turns sixteen.
Recommendation: Mechanisms should be put in place (in the form of a central website) that enable members of the public to check on the types of research being undertaken with Guthrie Card data and the outcomes.

---

incremental increases in use, beyond the approved types of health research. At the end of Day One, 10 out of 19 Jurors considered the use of newborn blood spots for research to be in the public interest with no concerns and 9 answered that they had minor concerns.

Day Two provided the opportunity for more focused, deliberative discussion on the considerations that Jurors felt were key in deciding whether research access to the newborn blood spots was in the public interest. Following further, more cautionary, witness sessions, Jurors remained generally positive in their views although there was more discussion about possible commercial access and how this might be controlled. Jurors' discussion moved from 'for' or 'against' arguments to more nuanced discussion in the deliberative phase. Jurors were asked to identify and agree on considerations they felt were key in deciding whether research access to the newborn blood spots was in the public interest. Five key considerations were identified, and a consensus statement achieved for the first four (Supplementary Table 2). The overall verdict of the Citizens' Jury was that access to the newborn blood spots heel prick samples would be in the public interest. The distribution of opinion recorded at the end of event questionnaire varied only slightly after the deliberation,

with 8 stating that they had no concerns, 9 that they had minor concerns and 1 who had major concerns. This verdict was accompanied by a clear articulation of four specific conditions and a fifth recommendation, as summarized in Table 2 and provided in full in the Supplementary Material.

**Stakeholder meeting and feasibility study approval**. The findings from the Citizens' Jury were shared at an independent Stakeholder Meeting convened by the Scottish Government's Chief Scientist Office (CSO) in January 2019. The meeting concluded that further work to explore feasibility should be conducted before wider public consultation. The proposed Scottish Government-led consultation would not only have to consider whether research access with suitable oversight and governance would be in the public interest, but also whether the proposed research was feasible and justified. Two categories of feasibility study were recommended, epidemiological and biological. A letter approving such a feasibility study was provided by the Chief Medical Officer for Scotland in September 2019. The Caldicott Guardians of NHS Greater Glasgow and Clyde and NHS Tayside granted approval in January 2020 and March 2020, respectively.

A substantial amendment to the Research Tissue Bank approval for Generation Scotland, a family and population-based cohort with consent to link to health records and conduct genetic studies, to cover the feasibility study was submitted to the Research Ethics Committee and approved in March 2020.

**Epidemiological potential**. We sought to evaluate the feasibility of deriving a population-level e-cohort from 1965 onwards though linkage of data held on the Scottish newborn blood spots archive to routine NHS Scotland health records. In the 1970's, Scotland progressively introduced the Community Health Index (CHI), a unique identifier for all persons born in Scotland or registered with a GP in Scotland[9,10]. Each individual's CHI is now associated with every NHS Scotland transaction. The CHI can be used to link all patient interactions, including primary, secondary and tertiary care, vaccination, laboratory results and prescriptions. All Scottish newborn blood spots have been catalogued from 2000 onwards, but not so for earlier cards. Indeed, there is no formal record of the way in which earlier cards were collected and archived. To establish the physical state of the cards and associated identifying information, NHS Greater Glasgow and Clyde Biorepository staff, who have Scotland-wide responsibility for the archive, inspected 30 boxes of newborn blood spots dating from 1965 to 1999 ('Methods', Table 3; Supplementary Figs. 1 and 2).

Each box contained circa 4000–6000 cards in bundles by area health board, main hospital and date. Boxes and bundles were visually inspected for their physical state and origin. The newborn blood spots archive is stored at ambient temperature, but in the early days there was no unified approach to the type of dried blood spot card used nor the storage conditions. Some batches of cards were visually damaged by damp and there was only partial correspondence between the box identifier and the contents ('Methods').

A random sub-set of newborn blood spots (circa 1% and circa 1200 in total) were inspected in more detail. Cards were checked for feasibility of linkage to NHS health records by noting whether the following types of information were included: child name, date and place of birth, hospital number, maternal and paternal name. Under the terms of the Caldicott Guardian's approval, only summary data was tabulated. No personal identifying information was collected or stored. We observed significant variation in the newborn blood spot information content, both over time and between different health boards. Despite these inconsistencies, for the great majority of newborn blood spots from 1965 onwards, there was sufficient information to provide direct or high-confidence probabilistic matching to CHI-linked records. This would provide a strong basis for anonymised, population-level epidemiological study, or of individual level research, subject to consent and other relevant permissions. That said, for the earlier cards to be systematically catalogued by time and birthplace, digitally searchable and linked to routine NHS health records would require substantial investment of time and effort.

**Biological potential**. Any proposed assay of the Scottish newborn blood spot archive of dried blood spots must take into consideration the finite nature of the resource, the scientific added value and public benefit gain. Pilot studies conducted between 2012 and 2014 on de-identified sample punches demonstrated that sufficient quantity and quality of DNA could be extracted for whole genome or exome sequencing from 128 of the 136 cards tested (94%) with closely comparable success rates between the earliest (1965) and the most recent (2012) cards sampled, and DNA yields (mean 19.9 μg, range 16.7–23.1 μg) ('Methods'; Table 1).

The next objective was to evaluate the feasibility of recovering epigenetic signatures from the Guthrie card blood spots of Generation Scotland volunteers through genome-wide DNA methylation array technology ('Methods'). This assay was selected because it consumes a very small amount of material (200–400 ng DNA) and would create a multipurpose dataset of potentially high value across a wide range of traits with predictive value[11,12].

Generation Scotland was the approved source of test samples for this purpose because volunteers gave broad consent for access to their medical records and to genetic and other biomarker studies[5,13]. Genome-wide DNA methylation data from peripheral blood DNA collected at recruitment baseline (2006–2011) was available for comparison and validation. Information on maternal smoking behaviour was also available as a test case for epigenetic validation using gene-specific and genome-wide scores previously linked to smoking[8].

Amongst the contents of the first 30 boxes examined for record linkage potential, we identified 92 newborn blood spots from consented members of the Generation Scotland research cohort. A condition of this feasibility study approval by the Caldicott Guardians was that cards could only be sampled if one of the four blood spots was left untouched. This condition meant that we were only able to take punches from 58 of the 92 cards retrieved (60%). DNA from these 58 punch samples (dating from 1983 to 1989) was extracted and a genome-wide DNA methylation study conducted ('Methods'). Results were compared to matched blood samples collected from the same volunteers at their baseline research clinic visit. There was excellent genetic congruence between the perinatal and adult samples (Supplementary Figs. 3 and 4, Supplementary Table 1).

In Generation Scotland, we have data collected at baseline on smoking behaviour. Using this information in combination with

---

**Table 3 Newborn blood spot archive examination and sample selection for analysis.**

Step 1 Information recorded (Boxes)
Box ID; Area Health board; Hospital; Type of card; Date of test; Child forename; Child surname; presence or absence of Community Health Index identifier.

Step 2 Information recorded (Cards)
Child DOB; Additional comments on card; Number of blood spots; Size of spots; Mother's CHI; Mother's forename, surname and birth name; Mother's date of birth; Address; Postcode; Whether the cards had been autoclaved prior to archiving; Any other comments.

Step 3 Sample selection
- Pseudonymised ID list of consented Generation Scotland (GS) participants provided to Greater Glasgow and Clyde Biorepository (GGCB)
- GGCB staff look for matches between newborn blood spots and GS IDs
- 92 matching cards identified and checked for usability
- 6–10 punches of 3 mm diameter taken from 58 cards and couriered to the Edinburgh Clinical Research Facility for DNA analysis

Step 4 Sample analysis
- DNA methylation data from newborn blood spots compared to DNAm in peripheral blood at time of recruitment (2006–2011)
- DNA methylation data from newborn blood spots analysed for smoking signatures and correlated with recorded maternal smoking status

genome-wide methylation data from newborn blood spots showed that DNA methylation signatures at birth reflected maternal smoking, replicating previous studies[7] (Fig. 1).

## Discussion

The Citizens' Jury strongly agreed that research access to the newborn blood spots was in the public interest. Yet, the articulation of key conditions provides clear evidence that careful and transparent regulation must be in place (Table 2). These results, based on in-depth, qualitative data, should inform future public engagement to ensure a wider mandate for research use of newborn blood spots. The findings of the Citizens' Jury illustrate how the public can engage in nuanced ways with the issue in hand and make recommendations relevant for policy. The findings, including the cautions and concerns, reflect other research that has explored public views on health data, health research, public benefits and data sharing and how these might be addressed[4,14,15]. That said, whereas Citizen Juries are carefully balanced to enable in-depth exploration of the nuances of individual perspectives, the modest number of Jurors means that they cannot fully represent the wider public. Wider public consultation, including but not restricted to a refined and expanded Citizens Jury, is recommended. This can now include discussion of feasibility and resourcing, which can further inform such citizen deliberations. We recommend a multi-method consultation to allow different modes of engagement.

Some of the suggested safeguards and other recommendations are covered under the respective responsibilities of the Caldicott Guardians, Public Benefit and Privacy Panel (part of the information governance structure in Scotland that scrutinizes requests for access to health and other personal data) and NHS Research Ethics Committees (who protect the rights, safety, dignity and wellbeing of participants and collectively safeguard and allow legitimate access to health data in the public interest), but others would require further action. In line with the Citizens' Jury, the Stakeholder Meeting recognised that a bespoke and dedicated governance and oversight structure would be advisable, at least in the development phase of any proposed lifting of the moratorium on research access, despite the range of protections already in place for the use of health data[16]. The consensus view from Stakeholders was that a system for "opt-out" would need to be developed in line with current developments in secondary use of health data, especially for studies linked to health records[16].

The follow-on feasibility study has established the potential value of the circa 3 million Scottish newborn blood spots archive for (a) population-level epidemiological studies, and (b) DNA-based biomarker studies of people born since 1965. This provides an evidence base for further public and stakeholder consultation on establishing a managed NHS resource for research.

In the process, our study did highlight practical issues regarding the currently uncatalogued, pre-2000 archive, constituting circa 2 million cards. Significant investment would be required to retrieve these boxes from long-term storage, catalogue their contents, render them searchable and return these to safe storage for future retrieval in an organized manner. Despite these caveats, the feasibility study demonstrated that the archive provided sufficient information for CHI seeding and linkage to NHS Scotland routine data sets. This could therefore provide the basis for population-level, epidemiological health data research. A fully catalogued, searchable dataset with independent oversight for regulated access and use, plus obligatory return of processed data and analysis would maximise the public benefit.

The feasibility study also demonstrated the potential value of the Scottish newborn blood spots archive for genetic (Table 1) and epigenetic studies (Fig. 1). DNA array technologies now require very small amounts of input material (200–400 ng), corresponding to less than one intact dried blood spot, and are readily scalable. If suitably indexed, a wide range of trait-defined or population-level studies would be possible. Indeed, we have shown that the epigenetic scores correlate well with circulating proteome scores and predict disease traits in Generation Scotland[17]. There is thus scope to investigate a wide range of health issues where ground-state knowledge at or around birth would be valuable. In most cases, the small amount of material required for DNA-based studies would leave material intact for other future assays. Again, obligatory return of processed data and analysis to the archive would maximise the public benefit. Retaining an archive of extracted DNA for additional and unforeseen uses would be prudent. These might include assays for the presence of non-host nucleic acid assays indicating prevalent infection. Feasibility studies to establish the possibility of other assays such as blood proteins, metabolites or environmental pollutants should be considered. In all cases, the public benefit would have to be seen to outweigh the cost and the consequential depletion of the archive.

Taken together, there is now an evidence base for wider public consultation to inform decisions about whether to lift the current research moratorium and if so under what ethics, oversight, governance and engagement terms and conditions. The public interest will however require that public support is established and maintained through transparency of purpose and demonstration of public benefit.

Our study has wider implications. Birth cohorts are a bedrock of much epidemiological research but are limited by cost, attrition and bias. Unbiased routine data collections have proved crucial in measuring the impact on health of environmental exposures, such as smoking, poisoning, pollution and infection, and policy changes introduced to mitigate these exposures. They do not however take account of biological factors. We now know that genetic factors influence resistance or vulnerability to all of these exposures to a greater or lesser extent. Population-level, newborn blood spot archives are uniquely positioned to integrate exposure data with biological data that is not compromised by selection bias. It would be possible to design well-powered studies against biological (including genetic) and non-biological (socio-demographic and environmental) criteria. The Scottish newborn blood spot archive has the potential to pioneer such an approach, subject to a positive outcome for the pending Public Consultation. Our study also provides a template for others to follow. In so doing, best practice could be shared, hypotheses tested, and published findings arising from conventional cohort studies replicated or refuted. For public support, robust and transparent governance must be put in place, with onward data sharing for legitimate use to maximise public benefit.

## Data availability

To access Generation Scotland data, including the data derived in the feasibility study described here, please go to www.ed.ac.uk/generation-scotland/for-researchers/access. Summary-level data analysed in this study (source data) has been deposited at https://www.github.com/marioni-group/guthrie.

## Code availability

Code from this study, along with version information on the software used, has been deposited here: https://doi.org/10.5281/zenodo.7043056[6].

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

## Acknowledgements

We thank Dr Tom Barlow, Scottish Government Chief Scientists Office for guidance and input into the feasibility studies described here; Dr Sarah Smith, Director of the National Screening Service for facilitating access to the Scottish newborn blood spot archive and sanctioning the work of the Greater Glasgow and Clyde Biorepository; Alexis Smith, Research Nurse at the Greater Glasgow and Clyde Biorepository; Caroline Clark, Mark Davidson and the NHS Grampian Genetics laboratory for technical support in the early feasibility studies, Dr Joan Mackenzie, NHS Scotland blood spot screening service and the Danish Newborn Screening Biorepository for technical advice and support for the pilot DNA sequencing of de-identified newborn blood spot punches. The Citizens' Jury was funded by a Wellcome Trust Institutional Science Support Fund award to the University of Edinburgh (S.C.B. and D.J.P.). Generation Scotland received core support from the Chief Scientist Office of the Scottish Government Health Directorates [CZD/16/6] and the Scottish Funding Council [HR03006] and is currently supported by the Wellcome Trust [216767/Z/19/Z] award to D.J.P. Baseline DNA methylation was funded by the Wellcome Trust (Wellcome Trust Strategic Award "STratifying Resilience and Depression Longitudinally" (STRADL) Reference 104036/Z/14/Z and Investigator Award 220857/Z/20/Z to A.Mc.I.). The feasibility study was supported in part by HDRI Health Data Research UK substantive hub award (EDIN1) and Chief Scientist Office Scotland (ETM19). C.H. and S.K. are supported by the Medical Research Council University Unit award to the MRC Human Genetics Unit, University of Edinburgh, grant number MC_UU_00007/10, Programme MC_PC_U127592696.

## Author contributions

Designed the study: D.J.P., S.C.B., J.P., Z.M., R.F., S.K., R.E.M. and A.C. Raised the funding: D.J.P., S.C.B., J.P., Z.M. and C.S. Designed and conducted the Citizens' Jury: D.J.P., S.C.B., C.P., M.A., C.M. and S.D. Conducted the examination of newborn blood spots: C.O. and S.Mc.C. Processed and analysed the newborn blood spots: K.W., L.M., N.W., S.Mc.C., D.Mc.C., R.E.M., K.W., D.St.C. and Z.M. Drafted the paper and prepared the study for submission and peer review: S.C.B., D.J.P. and D.Mc.C. Contributed text and/or figures: D.J.P., S.C.B., C.O., S.Mc.C., K.W., Z.M., D.Mc.C., R.E.M. and A.C. Reviewed drafts and approved the manuscript for submission: S.C.B., D.Mc.C., A.C., R.F., C.O., C.P., M.A., C.M., S.D., S.Mc.C., L.M., N.W., S.Mc.C., K.W., D.St.C., S.K., C.H., A.Mc.I., C.S., R.M., J.P., Z.M. and D.J.P.

## Competing interests

These authors declare the following competing interests: Ciaran Mulholland is a paid employee of Ipsos MORI. Sara Davidson is a paid employee of Ipsos MORI. Riccardo Marioni has received speaker fees from Illumina and is an advisor to the Epigenetic Clock Development Foundation. Lee Murphy has received payment from Illumina for presentations and consultancy. Andrew McIntosh has received research funding from the Sackler Trust and speaker fees from Janssen and Illumina. All other authors declare no competing interests.
