## [Peer Review File · Communications Medicine]

Reviewer #1 (Remarks to the Author):

Your manuscript explores the feasibility of sampling archived Guthrie newborn screening cards using a multi-pronged approach. You first explore the ethical, legal, and social issues using a citizen's jury and stakeholder meeting and then the technical feasibility of extracting DNA from the spots, usage for epigenomic analysis, and the reliability identifying individuals' samples.

The first phase of the project identified a useful and novel set of four specific conditions and a recommendation for ethical use of archived blood spots. The second phase highlighted some technical drawbacks, but you presented evidence that the majority of samples has sufficient DNA for epigenomic analysis.

Overall, this was a well written paper that was refreshingly broad in its scope. I expect it to be well cited because there is tremendous potential to unlock similar sets of samples around the world, especially those that have a current moratorium on sample usage.

You highlight some of the weaknesses of the study such as the relatively small scale of the citizens jury and the problems associated with long-term sample storage.

My main criticism is in the detail you provide; there isn't enough for the study to 'the ability of a researcher to reproduce the work, given the level of detail provided' as specified by the journal. I list the specifics and other minor criticisms below.

1. Please provide more (supplementary) details of the protocols used for the citizen's jury and stakeholder meeting
2. Please provide more (supplementary) details of the findings of the citizen's jury and stakeholder meeting
3. With regards to the recommendation in Table 1, was there any discussion related to an individual's right to access the data produced from research on Guthrie card DNA?
4. Lines 142-144: please state whether these boxes were selected at random or not?
5. Lines 175-177: 'This [methylation array] assay was selected because it consumes a very small amount of material and would create a multipurpose dataset of high value across a wide range of traits with predictive value'. There are no current predictive epigenetic tests that have undergone clinical trials so I would like you to tone this down a little; maybe 'potentially predictive'?
6. Please provide more details in the main text about the data presented in Figure 1 e.g. what is the EpiSmokEr score and how are the results similar to the those expected? Please also make the axis labels large enough to be read.
7. Lines 240-241 'DNA array technologies now require very small amounts of input material'. Please provide more information on what 'small' means.
8. I would also like more information on the QC of both the DNA extracted from the Guthrie spots (quantity and quality WRT low molecular weight DNA if possible) and the methylation array data (unusable probes or samples with low detection p-values).
9. As Table 3 is previously unpublished data, please provide more information, e.g. what does 'successful' mean and please supply qualitative and quantitative parameters and supplementary methods
10. Lines 400-40: please supply more details e.g. median yield
11. The data you used for the maternal smoking status was measured at the time of collection (birth). Do you see any drawback in using data from after pregnancy rather than pre-pregnancy?
12. Lines 424-426 "An individual's smoking status can be reliably predicted using composite DNA methylation-derived smoking scores, and effects have also been observed in the offspring of mothers who smoked during pregnancy." Please note that the data supports an association within the population rather than for individuals. Can you please amend?

Reviewer #2 (Remarks to the Author):

- An extremely well-researched, topical and timely paper
- Combines two public health interests: NBS and Epi data necessary for future public health strategies and implementation
- Well-written; chronology explicit; science rationale evident and methodology well-described
- Convincing? Yes. NBS's programmes are in newborns' immediate health interests and adult population cohorts, and the use of Guthrie cards is in adults' interests. NBS is a resource for more targeted personalized health efforts so that universal health care can be substantial and accessible for all actual and future citizens.
- This study is original, innovative, and promising (if approved) for other health care systems. Why did it take so long to do this?
- A table should clearly indicate that NBS will occur, so the "opt-in" is for de-identified research only.

Reviewer #3 (Remarks to the Author):

1. Thank you for the opportunity to review this fascinating study of real-life policy making on the use of Guthrie Cards. The paper describes a deliberative process, followed by two feasibility studies, to assess the potential for giving researchers access to Guthrie Card heel prick blood tests in Scotland. Overall, this is an important piece of work, but the way it is presented could be improved.

2. The introduction positions the activities reported in the paper in a specific political and social context. While that is appropriate for this study of evidence development in action, I think the introduction would benefit from framing the activities as a response to an overarching research question which might be something like 'under what conditions, if any, should researchers in Scotland be granted access to Guthrie Cards?' With such a question set out at the beginning of the paper, the combination of citizens' jury (to assess public support and identify conditions that could underpin such support), and the epidemiological and biological feasibility studies makes more sense.

3. There is now widespread access for research purposes to a huge amount of administrative health data, including in Scotland. The introduction would also benefit from explaining why a specific program of work has been needed to justify this access for Guthrie cards heel prick blood spots. Put another way, what makes the use of Guthrie Cards of particular social and ethical significance?

4. I found the combination of findings presented in the paper confusing. I was, in a sense, reading two papers, one presenting and analysing the outcomes of a citizens' jury on the social acceptability of research access to Guthrie Card heel prick blood tests and a second on the biological and epidemiological feasibility of using the Guthrie cards for research. The link between these two 'studies' seemed to be a stakeholder meeting but the logic of the link was not spelt out. The authors might find it helpful to reframe the Stakeholder Consultation more explicitly as part of the evidence generation. That would mean providing a description of how the consultation took place, the decisions made and the reasons for these decisions. Did that meeting accept the jury recommendations and conditions? In particular, what, if anything, was the relationship between the jury outcomes and the subsequent decision to undertake further work on the feasibility of research

access?

5. My question about the link between the jury and the feasibility studies is highlighted by the fact that I could not follow the logic of running the jury before the assessment of epidemiological and biological feasibility. Surely the assessment of feasibility would have provided relevant evidence for the jurors? For example, had the jury been aware that cataloguing the earliest Cards would require substantial time and effort they might have made recommendations about limiting the date before which cards would not be accessed. If the order of events is simply just 'how things turned out' it would be helpful to understand why these events happened in the order they did and whether the authors actually think this was the most appropriate way for the activities to be completed.

6. The rationale for the choice of question/charge for the jury is not clear to me. Serving the public interest is often a necessary, but not sufficient, reason for allowing a specific activity to go ahead. It therefore seems possible that the jury could say 'yes' to the question and also say that research access should not be granted. In fact, the conditions and recommendation the jury made seem to actually answer the question 'Under what conditions should researchers be granted access to the Guthrie Card heel prick blood tests?' rather better than they do the question actually used. Of course, the jury question cannot be changed at this stage, so I would like to understand how this question was settled on and, particularly, whether the jury were actually responding to a more general question such as the one I suggest above.

7. The outcomes of the jury seem to be presented as two separate days and I would encourage the authors to revisit this way of presenting the findings. The outcomes of citizens' juries are based on the total body of evidence the jurors hear and deliberate on, in this case, over two full days. It is therefore the recommendations and conditions which are of greatest significance. I think it would make more sense to describe the final recommendation and conditions in more detail, including the jurors' reasons, rather than to recount the jurors' discussion on the way through. What particular logics and reasons underpinned the recommendations and conditions? What sorts of reasons mattered the most? Which factors were of lesser importance?

8. The discussion recommends that wider public consultation should occur. I would like to see the authors address the question of what forms further public consultation might take and why it should be undertaken. For example, it is well recognised that deliberative studies tend to lead to greater support for use of public data than surveys. A survey of the Scottish public might well find less support for research access to Guthrie Cards. What would this mean, particularly in light of the earlier jury outcomes?

9. As a minor note, the sentence 'Following on from the CJ....studies for the use of health data' belongs in the body of the text, not the discussion.

10. Finally, the wider implications section seems to assume that the Scottish Guthrie card archive will inevitably go ahead, in some form. That may be true, but it is not actually an implication of this study. I would be looking to read something about the actual study undertaken – which concerns a specific way of assessing the potential for use of sensitive health data.

Reviewer #4 (Remarks to the Author):

Dear Sarah and all

Paper Overview

The Scottish government commissioned a public consultation and deliberation process as to whether the newborn Guthrie blood spot cards could form the basis of a national birth cohort study. This paper reports on both a Citizen's Jury that considered the issues raised and an assessment of the state of the Guthrie bloodspot cards collected since 1965, confirming the feasibility of such an epidemiological study with linkage to other NHS Scotland data on the patients. Such a study would be internationally important, being much larger in scale than ALSPACH (for example). Furthermore, the samples of 58 participants in the Generation Scotland genome study were shown to yield DNA of sufficient quality for genomic, epigenomic and protein-based analyses, although a substantial commitment would be required to make pre-2000 Guthrie cards fully available to research.

Assessment

The paper is clearly written and tells an interesting story. It is virtually free of typos. If the Scottish Guthrie cards do become the basis for a national birth cohort study, this preliminary work will become important and will need to be documented, as has been done here. I have no substantive criticisms of the paper but wonder if all of the detail included is of great interest, until it is known whether the Scottish national birth cohort study is indeed established. Whereas the information about the citizens' jury and the outline information about the Guthrie card storage and the feasibility of Omics studies is of general interest, I wondered if all the technical information would be needed if the epidemiological study does not materialise. If the editors suggest revision, then the extent of this detail could perhaps be trimmed back.

Typos etc

Page 4, para 1, line 18: delete 'or not' — and elsewhere in paper (e.g. page 4, para 2, lines 2 and 8)

Page 5 para 2 line 5: that => whether

Page 8 para 2 line 6: others. => others (i.e. delete '.')

Regards,

Angus Clarke

2022-06-01 Response to reviewers

To the editor and reviewers:

We thank the editors and reviewers for the time and care they have taken to comment on our study. We are delighted by their positive and constructive remarks, and this opportunity to respond. As requested, we have provided the editors with a (yellow) highlighted revised version. There are no changes to the authorship, but we have taken this opportunity to correct one or two contact details (revised main text and separate document). Similarly, we have added further information on the ethics approval process for the Citizens Jury.

In response to Reviewer 1, we have attached a text edited, but otherwise complete version of the Ipsos MORI report on the Citizens Jury as separate file (attached). We wish to include this as Supplementary Information. We have also added information (highlighted in yellow, lines 556-568) on the ethics approval for the Citizens Jury which was inadvertently missing from the original submission.

When we submitted this paper, the Scottish Government CSO used the terminology 'Scottish Guthrie card archive' but have since adopted the more universally recognised form 'Scottish newborn blood spot archive', which we have adopted.

Changes made to the text and figures in response to reviewers' comments are also highlighted in yellow.

For ease of navigation, the revised version has line numbers, and these are referred to in our response where appropriate. Substantive changes are pasted into the response where appropriate.

Response to Reviewers (embedded)

Reviewer #1 (Remarks to the Author):

Your manuscript explores the feasibility of sampling archived Guthrie newborn screening cards using a multi-pronged approach. You first explore the ethical, legal, and social issues using a citizen's jury and stakeholder meeting and then the technical feasibility of extracting DNA from the spots, usage for epigenomic analysis, and the reliability identifying individuals' samples.

The first phase of the project identified a useful and novel set of four specific conditions and a recommendation for ethical use of archived blood spots. The second phase highlighted some technical drawbacks, but you presented evidence that the majority of samples has sufficient DNA for epigenomic analysis.

Overall, this was a well written paper that was refreshingly broad in its scope. I expect it to be well cited because there is tremendous potential to unlock similar sets of samples around the world, especially those that have a current moratorium on sample usage.

You highlight some of the weaknesses of the study such as the relatively small scale of the citizens jury and the problems associated with long-term sample storage.

My main criticism is in the detail you provide; there isn't enough for the study to 'the ability of a researcher to reproduce the work, given the level of detail provided' as specified by the journal. I list the specifics and other minor criticisms below.

1. Please provide more (supplementary) details of the protocols used for the citizen's jury and stakeholder meeting

Response: We have attached a text edited, but otherwise complete version of the Ipsos MORI report on the Citizens Jury as separate file (attached). We wish to include this as Supplementary Information. We cannot provide full details of the Stakeholder Meeting as this is owned by the Scottish Government. The account provided here has been approved, but it is their decision as to when and how this will be published in full.

2. Please provide more (supplementary) details of the findings of the citizen's jury and stakeholder meeting.

Response: Please see response above to Point 1.

3. With regards to the recommendation in Table 1, was there any discussion related to an individual's right to access the data produced from research on Guthrie card DNA?

Response: Not specifically, but there was an implicit understanding that research access could lead to novel and improved clinical tests.

4. Lines 142-144: please state whether these boxes were selected at random or not?

Response: the text (Lines 388-391) has been amended to clarify this.

The set of 30 boxes retrieved and documented for epidemiological purposes were selected on the basis that a) Greater Glasgow and Tayside were the regions for which we had Caldicott Guardian approval and b) they were expected to include bundles from Generation Scotland participants as the majority came for these regions.

5. Lines 175-177: 'This [methylation array] assay was selected because it consumes a very small amount of material and would create a multipurpose dataset of high value across a wide range of traits with predictive value'. There are no current predictive epigenetic tests that have undergone clinical trials so I would like you to tone this down a little; maybe 'potentially predictive'?

Response: the text has been modified accordingly (Lines 242-254 and 253-254).

6. Please provide more details in the main text about the data presented in Figure 1 e.g. what is the EpiSmokEr score and how are the results similar to the those expected? Please also make the axis labels large enough to be read.

Response: Figure 1 and legend have been revised in response (lines 609-618).

Figure 1: EpiSmokEr score (A) and AHR DNAm (B) plotted against maternal smoking status

Upper and lower hinges correspond to the upper and lower quartiles, respectively. Whiskers extend to data points as far as 1.5 times the interquartile range. Outlying data points are defined as those beyond the whiskers. Thick horizontal lines represent the median.

7. Lines 240-241 'DNA array technologies now require very small amounts of input material'. Please provide more information on what 'small' means.

Response: the text has been modified accordingly (Lines 242-254 and 253-254).

8. I would also like more information on the QC of both the DNA extracted from the Guthrie spots (quantity and quality WRT low molecular weight DNA if possible) and the methylation array data (unusable probes or samples with low detection p-values).

Response: the text and methods sections have been modified accordingly (lines 434-437).

Quality control measures were performed, removing probes with high detection p-values (>0.05) in $>5\%$ of samples ($N=52,375$), or a beadcount <3 in more than 5% of samples ($N=5,038$). Three samples were removed for having $>5\%$ of sites with a detection p-value >0.05 .

9. As Table 3 is previously unpublished data, please provide more information, e.g. what does 'successful' mean and please supply qualitative and quantitative parameters and supplementary methods

Response: we have added a further column to Table 3 and the text and methods sections have been modified accordingly.

Time period	New-born blood spots sampled	Successful* DNA extractions	Extractions that failed DNA sequencing	Mean yield of DNA extracted (µg)	Net yield for analysis
1965-1974	17	15	0	16.7	88%
1975-1984	32	31	1	19.5	94%
1985-1994	17	16	1	17.3	88%
1995-2004	18	17	0	23.1	94%
2005-2019	16	16	1	22.9	94%
2010-2013	36	36	0	19.7	100%
Sum	136	131	3	19.9	94%

*In 5 / 136 (3.8%) the extraction process failed to recover measurable quantities of DNA using the methods described.

10. Lines 400-40: please supply more details e.g. median yield

Response: see modified Table above (Point 9) and Lines 414-418.

Exome sequencing used the Ion AmpliSeq exome kit run on IonTorrent Proton sequencer. Data analysis and variant calling used the IonReporter IonExpress variant caller, 42 – 45 million mapped reads, 94.2 – 94.7% on target, mean depth 122 – 130 reads per sample. Thirty one of 32 runs met standard QC criteria for variant analysis.

11. The data you used for the maternal smoking status was measured at the time of collection (birth). Do you see any drawback in using data from after pregnancy rather than pre-pregnancy?

Response: Ideally, smoking status would be established by direct assay (cotinine, or similar) before and during pregnancy, but that was not possible. Maternal smoking status before and during pregnancy was by self-report and collected by questionnaire at baseline (2006-2011). See modified text Lines 456-459.

Maternal smoking status at the time of sample collection was derived from smoking status at GS baseline, and the “years stopped” variable for former smokers, where both mother and baby are in GS. DNA methylation-based estimates of smoking status were obtained, using previously validated methods^{11,12}.

12. Lines 426-428 “An individual’s smoking status can be reliably predicted using composite DNA methylation-derived smoking scores, and effects have also been observed in the offspring of mothers who smoked during pregnancy.” Please note that the data supports an association within the population rather than for individuals. Can you please amend?

Response: You are of course correct and the text has been amended to say so (Line 462).

Reviewer #2 (Remarks to the Author):

- An extremely well-researched, topical and timely paper
- Combines two public health interests: NBS and Epi data necessary for future public health strategies and implementation
- Well-written; chronology explicit; science rationale evident and methodology well-described
- Convincing? Yes. NBS's programmes are in newborns' immediate health interests and adult population cohorts, and the use of Guthrie cards is in adults' interests. NBS is a resource for more targeted personalized health efforts so that universal health care can be substantial and accessible for all actual and future citizens.
- This study is original, innovative, and promising (if approved) for other health care systems. Why did it take so long to do this?

Response: This study reflects over 10 years of effort by us and others to address the question. The moratorium on research access was applied UK-wide under the joint authority of the Chief Medical Officers for England, Wales and Northern Ireland. This was at a time of sensitivity around genetic testing and consent. A Westminster Government led, UK-wide consultation was promised, but never materialised. The Scottish Government proactively commissioned the 2013 ELSI report widely referred to. As academics, we took the initiative to conduct a Citizens Jury to test the public attitude to the question of public benefit from research access. Based on both the ELSI report and the Citizens Jury, plus the Scottish Government led Stakeholders Meeting, the CMO for Scotland approved the partial lifting of the moratorium for the study described here. The political process took time. The approval coincided with the Covid-19 pandemic. It all added up to a very extended period.

- A table should clearly indicate that NBS will occur, so the "opt-in" is for de-identified research only.

Response: We take your point, but the Table is taken verbatim from the Citizens Jury report, so cannot be amended. The Citizens' Jury focussed on research access and this has been reinforced in the title of Table 1 (Line 597).

Table 1: Citizens Jury conditions and recommendations regarding research access

Reviewer #3 (Remarks to the Author):

1. Thank you for the opportunity to review this fascinating study of real-life policy making on the use of Guthrie Cards. The paper describes a deliberative process, followed by two feasibility studies, to assess the potential for giving researchers access to Guthrie Card heel prick blood tests in Scotland. Overall, this is an important piece of work, but the way it is presented could be improved.

2. The introduction positions the activities reported in the paper in a specific political and social context. While that is appropriate for this study of evidence development in action, I think the introduction would benefit from framing the activities as a response to an overarching research question which might be something like 'under what conditions, if any, should researchers in Scotland be granted access to Guthrie Cards?' With such a question set out at the beginning of the

paper, the combination of citizens' jury (to assess public support and identify conditions that could underpin such support), and the epidemiological and biological feasibility studies makes more sense.

Response: We take your point and have added an introductory section to the main text making that overall aim clearer (Lines 65-70).

Longitudinal population health studies with extensive record linkage and deep genomic annotation have the potential to realise the objectives of preventative and predictive medicine. Ideally, these studies should cover the full life course. Epidemiological and biological analyses of new-born blood spots have the potential to anchor longitudinal studies at birth. However, issues of consent and sensitivity over genetic studies first need careful consideration.

We were constrained by several limitations. This study reflects over 10 years of effort by us and others to address the overall question of research access to the new-born blood spot archive in Scotland. The moratorium on research access was applied UK-wide under the joint authority of the Chief Medical Officers for England, Wales and Northern Ireland. This was at a time of sensitivity around genetic testing and consent. A Westminster Government led, UK-wide consultation was promised, but never materialised. The Scottish Government proactively commissioned the 2013 ELSI report widely referred to. As academics, we took the initiative to conduct a Citizens Jury to test the public attitude to the question of public benefit from research access. Based on both the ELSI report and the Citizens Jury, plus the Scottish Government led Stakeholders Meeting, the CMO for Scotland approved the partial lifting of the moratorium for the study described here. This study went as far as we were allowed. It will feed into a Public Consultation that will be devised and undertaken by the Scottish Government.

3. There is now widespread access for research purposes to a huge amount of administrative health data, including in Scotland. The introduction would also benefit from explaining why a specific program of work has been needed to justify this access for Guthrie cards heel prick blood spots. Put another way, what makes the use of Guthrie Cards of particular social and ethical significance?

Response: We have touched upon some of the issues in our response to your Point 2 and have added text pointing out the sensitivity of genetic data in particular. These issues were considered in depth by the 2013 CSO ELSI report. We have attached a text edited, but otherwise complete version of the Ipsos MORI report on the Citizens Jury as separate file (attached). We wish to include this as Supplementary Information. We have heeded your comments and modified the text as best we can within these constraints.

4. I found the combination of findings presented in the paper confusing. I was, in a sense, reading two papers, one presenting and analysing the outcomes of a citizens' jury on the social acceptability of research access to Guthrie Card heel prick blood tests and a second on the biological and epidemiological feasibility of using the Guthrie cards for research. The link between these two 'studies' seemed to be a stakeholder meeting but the logic of the link was not spelt out.

The authors might find it helpful to reframe the Stakeholder Consultation more explicitly as part of the evidence generation. That would mean providing a description of how the consultation took place, the decisions made and the reasons for these decisions. Did that meeting accept the jury recommendations and conditions? In particular, what, if anything, was the relationship between the

jury outcomes and the subsequent decision to undertake further work on the feasibility of research access?

Response: You have highlighted both the strengths (a combined public consultation and feasibility study) and weaknesses (or rather constraints) to this study. See Response to Reviewer 1, Point 1, Reviewer 2, and to your Points 2 and 3. We have made some changes to the text as a best attempt to respond to your sound criticisms with the constraints placed on us by an ongoing, Scottish Government led process. As a key part of a long process of evidence building to inform policy, the paper intentionally combines approaches to provide context for each.

5. My question about the link between the jury and the feasibility studies is highlighted by the fact that I could not follow the logic of running the jury before the assessment of epidemiological and biological feasibility. Surely the assessment of feasibility would have provided relevant evidence for the jurors? For example, had the jury been aware that cataloguing the earliest Cards would require substantial time and effort they might have made recommendations about limiting the date before which cards would not be accessed. If the order of events is simply just 'how things turned out' it would be helpful to understand why these events happened in the order they did and whether the authors actually think this was the most appropriate way for the activities to be completed.

Response: We agree, but for the reasons discussed above, we can only report 'how things turned out'. Ideally, a programme of public consultation and a second Citizens Jury's should follow. We have modified the 'limitations' to reflect your important point (Lines 220-223).

Wider public consultation, including but not restricted to a refined and expanded Citizens Jury, is recommended. This can now include discussion of feasibility and resourcing, which can further inform such citizen deliberations. We recommend a multi-method consultation to allow different modes of engagement.

6. The rationale for the choice of question/charge for the jury is not clear to me. Serving the public interest is often a necessary, but not sufficient, reason for allowing a specific activity to go ahead. It therefore seems possible that the jury could say 'yes' to the question and also say that research access should not be granted. In fact, the conditions and recommendation the jury made seem to actually answer the question 'Under what conditions should researchers be granted access to the Guthrie Card heel prick blood tests?' rather better than they do the question actually used. Of course, the jury question cannot be changed at this stage, so I would like to understand how this question was settled on and, particularly, whether the jury were actually responding to a more general question such as the one I suggest above.

Response: These are all good points and ones which could be explored in the future. If the editors agree to include the full Ipsos MORI report as Supplementary text, that will go some way towards addressing your comments, not least setting the specific question on new-born blood spots within the context of the state-of-the-art in genome technology and genetic epidemiology in 2017. The concept of public interest is key to research governance. This is what we wanted the Citizens Jury to deliberate. The Jurors moved towards conditionality as they explored the issues around the research use of new-born blood spots.

7. The outcomes of the jury seem to be presented as two separate days and I would encourage the

authors to revisit this way of presenting the findings. The outcomes of citizens' juries are based on the total body of evidence the jurors hear and deliberate on, in this case, over two full days. It is therefore the recommendations and conditions which are of greatest significance. I think it would make more sense to describe the final recommendation and conditions in more detail, including the jurors' reasons, rather than to recount the jurors' discussion on the way through. What particular logics and reasons underpinned the recommendations and conditions? What sorts of reasons mattered the most? Which factors were of lesser importance?

Response: Again, these are all good points which would be addressed in large part by the inclusion of the full Ipsos MORI report as Supplementary Material. We have also revised the main text to address these points. The analysis of the two days demonstrates the generative nature of the conditions and follows the arc of the deliberative process as the jurors gradually move towards recommendations having considered evidence and debated issues.

8. The discussion recommends that wider public consultation should occur. I would like to see the authors address the question of what forms further public consultation might take and why it should be undertaken. For example, it is well recognised that deliberative studies tend to lead to greater support for use of public data than surveys. A survey of the Scottish public might well find less support for research access to Guthrie Cards. What would this mean, particularly in light of the earlier jury outcomes?

Response: We have revisited this aspect of the study and made our own suggestions, within the constraints imposed by the on-going, parallel, but independent plans by the Scottish Government, including the need for a multi-method approach..

9. As a minor note, the sentence 'Following on from the CJ....studies for the use of health data' belongs in the body of the text, not the discussion.

Response: This has been addressed (Line 230).

10. Finally, the wider implications section seems to assume that the Scottish Guthrie card archive will inevitably go ahead, in some form. That may be true, but it is not actually an implication of this study. I would be looking to read something about the actual study undertaken – which concerns a specific way of assessing the potential for use of sensitive health data.

Response: Points taken and responded to (Lines 283-284).

subject to a positive outcome for the pending Public Consultation. Our study also provides a template for others to follow.

Reviewer #4 (Remarks to the Author):

Paper Overview

The Scottish government commissioned a public consultation and deliberation process as to whether the newborn Guthrie blood spot cards could form the basis of a national birth cohort study. This paper reports on both a Citizen's Jury that considered the issues raised and an assessment of the state of the Guthrie bloodspot cards collected since 1965, confirming the feasibility of such an epidemiological study with linkage to other NHS Scotland data on the patients. Such a study would be internationally important, being much larger in scale than ALSPACH (for example). Furthermore,

the samples of 58 participants in the Generation Scotland genome study were shown to yield DNA of sufficient quality for genomic, epigenomic and protein-based analyses, although a substantial commitment would be required to make pre-2000 Guthrie cards fully available to research.

Assessment

The paper is clearly written and tells an interesting story. It is virtually free of typos. If the Scottish Guthrie cards do become the basis for a national birth cohort study, this preliminary work will become important and will need to be documented, as has been done here. I have no substantive criticisms of the paper but wonder if all of the detail included is of great interest, until it is known whether the Scottish national birth cohort study is indeed established. Whereas the information about the citizens' jury and the outline information about the Guthrie card storage and the feasibility of Omics studies is of general interest, I wondered if all the technical information would be needed if the epidemiological study does not materialise. If the editors suggest revision, then the extent of this detail could perhaps be trimmed back.

Typos etc

Page 4, para 1, line 18: delete 'or not' — and elsewhere in paper (e.g. page 4, para 2, lines 2 and 8)

Page 5 para 2 line 5: that => whether

Page 8 para 2 line 6: others. => others (i.e. delete '.')

Response: On your point about whether 'all of the detail is of great interest', we have been explicitly asked for this detail as part of the approval for the feasibility study and other reviewers have asked for more, so the balance lies on inclusion, supported by well-considered peer review. Thank you for noting the typographical errors. These have been corrected.

REVIEWERS' COMMENTS:

Reviewer #1 (Remarks to the Author):

I am satisfied that you have addressed all my comments.

Reviewer #3 (Remarks to the Author):

Thank you for your very helpful explanations and responses to all of the reviewer comments. It helps a great deal to have some of the background and context for the research. The addition of the full Ipsos Mori report as a Supplementary file will be very helpful in this respect.